# Quality of life and well-being problems in secondary schoolgirls in Kenya: Prevalence, associated characteristics, and course predictors

Philip Spinhoven[1,2], Garazi Zulaika[3], Elizabeth Nyothach[4], Anna Maria van Eijk[3], David Obor[4], Eunice Fwaya[5], Linda Mason[3], Duolao Wang[3], Daniel Kwaro[4], Penelope A. Phillips-Howard[3]*

1 Institute of Psychology, Leiden University, Leiden, The Netherlands, 2 Department of Psychiatry, Leiden University Medical Center, Leiden, The Netherlands, 3 Liverpool School of Tropical Medicine (LSTM), Liverpool, United Kingdom, 4 Centre for Global Health, Kenya Medical Research Institute (KEMRI), Kisumu, Kenya, 5 Ministry of Health, Siaya County, Kenya

* Penelope.Phillips-Howard@lstmed.ac.uk

**Data Availability Statement:** This study was conducted with approval from the Kenya Medical Research Institute (KEMRI) Scientific and Ethics

## Abstract

### Background

Adolescents in sub-Saharan Africa often report low levels of quality of life (QoL) and well-being, but reliable data are limited. This study examines which sociodemographic, health, and behavioral risk factors and adverse adolescent experiences are associated with, and predictive of, QoL in Kenyan secondary schoolgirls.

### Methods and findings

3,998 girls at baseline in a randomised controlled trial in Siaya County, western Kenya were median age 17.1 years. Subjectively perceived physical, emotional, social and school functioning was assessed using the Pediatric Quality of Life (QoL) Inventory-23. Laboratory-confirmed and survey data were utilized to assess sociodemographic, health and behavioral characteristics, and adverse adolescent experiences. We identified a group of girls with Low QoL (n = 1126; 28.2%), Average QoL (n = 1445; 36.1%); and High QoL (n = 1427; 35.7%). Significantly higher scores on all well-being indicators in the LQoL compared with HQoL group indicated good construct validity (Odds Ratio's (ORs) varying from 3.31 (95% CI:2.41–4.54, p < .001) for feeling unhappy at home to 11.88 (95%CI:7.96–17.74, p< .001) for PHQ9 defined possible caseness (probable diagnosis) of depression. Adverse adolescent experiences were independently statistically significant in the LQoL compared to the HQoL group for threats of family being hurt (aOR = 1.35,1.08–1.68, p = .008), sexual harassment out of school (aOR = 2.17,1.79–2.64, p < .001), and for menstrual problems like unavailability of sanitary pads (aOR = 1.23,1.05–1.44, p = .008) and stopping activities due to menstruation (aOR = 1.77,1.41–2.24, p < .001). After 2-years follow-up of 906 girls in the LQoL group, 22.7% persisted with LQoL. Forced sex (aOR = 1.56,1.05–2.32, p = .028) and

Review Unit (SERU), which requires that data be released from any KEMRI-based Kenyan studies (including de-identified data) only after their written approval for additional analyses. In accordance, data for this study will be available upon request, after obtaining written approval for the proposedanalysis from the KEMRI SERU. Their application forms and guidelines can be accessed at https://www.kemri.org/seru-overview. To request these data, please contact the KEMRI SERU at seru@kemri.org.

**Funding:** This work is part of the Cups or Cash for Girls randomised-controlled trial funded by the Joint Global Health Trials Initiative (UK-Medical Research Council (MRC), the Foreign, Commonwealth & Development Office (FCDO), Wellcome, and the Department of Health and Social Care (DHSC) – https://www.ukri.org/opportunity/joint-global-health-trials/). PPH, DK, and DW received the award, grant MR/N006046/1. The funders had no role in study design, data collection and analysis, decision to publish, or preparation of the manuscript.

**Competing interests:** The authors have declared that no competing interests exist.

threats of family being hurt (aOR = 1.98,1.38–2.82, p < .001) were independent predictors of persistent LQoL problems.

## Conclusions

Persistent QoL problems in Kenyan adolescent girls are associated with adverse physical, sexual and emotional experiences and problems with coping with their monthly menstruation. A multi-factorial integral approach to reduce the rate of adverse adolescent experiences is needed, including provision of menstrual hygiene products.

## Trial registration

ClinicalTrials.gov:NCT03051789.

## Introduction

Common mental disorders (CMDs) such as depression, anxiety, and post-traumatic stress disorder (PTSD) are prevalent among children and adolescents worldwide, with an estimated prevalence of 25–31% [1]. In sub-Saharan Africa, 23% of the population comprises adolescents between 10 and 19 years of age [2]. The literature to date suggests that the proportion of disease burden in children and youth attributable to mental and substance use disorders is smaller in low-and middle-income countries (LMIC) due to higher rates of other ongoing diseases. This burden will increase as reproductive health and the management of infectious diseases improve in LMICs [3]. Despite this, reliable data on the prevalence and associated characteristics and risk factors of adolescent CMDs in sub-Saharan Africa are limited.

A recent systematic review examined the prevalence rate of child and adolescent mental health problems in sub-Saharan across 37 studies published since 2008, spanning 97,616 adolescents in the general population. The median point prevalence rates were 26.9% (IQR:20.1–31.1) for depression, 29.8% (IQR:18.6–36.65) for anxiety disorders, 40.8% (IQR:31.2–41.4) for emotional and behavioral problems, 21.5% (in only one study) for PTSD, and 20.8% (IQR:13.2–23.6) for suicidal ideation [4].

Studies included in a previous systematic review [5] and the review mentioned above [4] identified a number of factors that contribute to adolescent mental health morbidity, including a combination of sociodemographic factors (such as female gender, older age, impoverished living conditions, belonging to a minority tribe, being out-of-school, orphanhood, a large number of siblings, married or promised in marriage), health factors (such as HIV/AIDS, substance/alcohol abuse, low physical activity) and adverse adolescent experiences (such as exposure to violence, forced or unsafe sex, a recent change in residency, and maladaptive parental behaviour).

In Kenya, a few studies have focused on mental health in adolescents in normative environments such as secondary schools [6–11]. Prevalence rates of anxiety and depression symptoms and syndromes were elevated but varied between studies depending on the instruments and cut-off points used. Most studies reported more emotional and behavioral problems in girls compared to boys [6,8–11]. Relationships with older age [1,8], overcontrolling parental behaviour [7,9], being on a boarding school [9], and belonging to a minority tribe were found [10].

Mental health is more than the absence of mental disorders. Individuals' view of their own quality of life (QoL) offers a broader conceptualization of health and may yield a useful

additional indicator of mental health [12]. QoL encompasses a subjective evaluation of an individual's well-being as well as objective descriptions of individuals and their circumstances that are associated with subjective well-being. This study is restricted to an evaluation of subjective child and adolescent QoL, comprising subjective dimensions around participants' experiences of physical, psychological, social, and school functioning [12]. As far as we know, only two studies have assessed QoL in children and adolescents in sub-Saharan Africa [12,13]. Both studies reported relatively high levels of QoL and noted significant associations between QoL and mental health problems. Girls reported lower levels of QoL on all dimensions compared with boys [12]. The lower QoL in girls may be due to exposure to sexual and reproductive health harms, which are disproportionately high among adolescent girls in sub-Saharan Africa [14].

Although the number of epidemiological studies examining child and adolescent QoL problems in sub-Saharan Africa is increasing, there is a dearth of longitudinal studies examining the extent to which sociodemographic, psychosocial, health and behavioral factors play a role in the onset and maintenance of these problems. The aim of this study was to assess the prevalence, associated characteristics and predictive factors of QoL among schoolgirls in Kenya. It was hypothesized that a substantial subgroup of schoolgirls would report a low QoL, that schoolgirls with a low QoL would report more adverse adolescent experiences and sexual and reproductive health issues and that these negative experiences and health issues would be predictive of persistent low QoL.

## Materials and methods

### Study design and participant recruitment

The present study is nested in the Cups or Cash for Girls (CCG) cluster randomized controlled trial in rural western Kenya. CCG was a 4-arm trial evaluating the effect of menstrual cups, conditional cash transfer, a combination of both interventions, against controls on a composite of deleterious outcomes, [14] ClinicalTrials.gov NCT03051789. This trial offers a unique opportunity to focus specifically on the importance of sexual and reproductive health (SRH) issues for QoL among adolescent secondary school girls in Kenya. This study presents cross-sectional baseline data collected at enrolment for the CCG Trial and longitudinal follow-up data at 1- and 2-year follow-up (see [14] for a more detailed description of study aims, recruitment and procedures).

### Study area and population

This study was conducted in 96 rural or peri-urban secondary day schools spread across an area of approximately 2,500 km$^2$ in Siaya County (comprising the sub-counties of Gem, Siaya, Rarieda, Ugenya, and Ugunja sub-counties) in western Kenya. Lake Victoria is on its southern boarder, and the area is about 40 km from Kisumu City. Siaya's health profile is similar to relatively poor rural areas across Africa, with high endemicity of malaria, HIV, TB, and schistosomiasis close to the lake shores [15].

The CCG Trial targeted girls attending the selected study schools. These schools were eligible for selection if they taught non-boarding female day scholars and had permission from the school principals to participate. They were excluded if they catered for special needs students (i.e., schools for the blind), were for males only, or only accepted full board students. Participants were eligible if they were female, resided in the area, attended an eligible study school, were day scholars in the designated class years at enrolment, had informed parent–guardian consent, gave their individual informed assent, had reached menarche, were not visibly or declared pregnant at enrolment, and had no disability precluding participation.

## Outcome measures

The main outcome variable for girls' QoL was the Pediatric Quality of Life Inventory—23 items (PedsQL-23) [16]. The PedsQL uses 23 individual items to rate girls' well-being along four dimensions: physical, emotional, social, and school well-being. The internal consistency of the PedsQL-23 subscales across time in the present study was satisfactory to good: median 0.81 (range 0.68–0.89). Internal consistency of the total scale was excellent: baseline: 0.91; follow up after one year (FU1): 0.92; and follow up after 2 years (FU2): 0.94. In order to assess the construct validity of the QoL dimensions the following measures were selected: the EuroQol five-dimensional three-levels questionnaire (EQ-5D-3L), a descriptive system comprising the following five dimensions: mobility, self-care, usual activities, pain/discomfort and anxiety/depression [17]; two single item questions on feeling happy at school (yes/no) and feeling happy at home (yes/no); and the Patient Health Questionnaire-9 (PHQ-9) (only administered at the annual two follow-ups). The PHQ-9 is a self-report scale, which scores each of the 9 DSM-IV criteria for depression as "0" (not at all) to "3" (nearly every day) [18]. The internal consistency of the PHQ-9 in the present study was satisfactory to good: 0.79 at FU1 and 0.81 and FU2. Caseness is the terminology used to describe the probable diagnosis of depression based on PHQ-9 responses.

## Risk factors

This post hoc exploratory study utilized the same risk factors presented in the CCG trial baseline analysis of factors associated with the prevalence of HIV, HSV-2, pregnancy, and reported sexual activity [19]. Risk factors explored included individual characteristics (age, body mass index [BMI], and sexual activity), family characteristics (marital status, caring for a baby at home, and having no living parent), household characteristics (socio-economic status (SES)), lifestyle characteristics (self-reported alcohol use, smoking, and working outside of school and home), sexual harassment (at school, and out of school), menstruation-related characteristics (early menarche, using sanitary pads for menstruation, menstrual related school absence, stopping daily activities due to menstruation, severity of menstruation, duration of bleeding, and having to do activities such as housework, childcare, or offering sexual favours to obtain menstrual pads), and financial characteristics (source of money and transactional sex). In addition, outcomes in the CCG baseline analysis [19] were included as predictors (i.e., HIV and HSV-2 infections, and history of pregnancy). Survey data related to exposure to physical abuse (physical assault, being robbed, threats to hurt you, threats for family to be hurt) and emotional abuse (being publicly humiliated) were also added as putative risk factors for QoL.

## Statistical analyses

**QoL subgroups, prevalence and course.**   To reduce the number of QoL measures while keeping a comprehensive and detailed view of QoL, Latent Class Analysis (LCA) and Latent Transition Analysis (LTA) with continuous observed variables were used [20]. LCA is a person-centered approach, used to cluster participants rather than variables allowing identification of latent subgroups within the data. In the present study each latent class represents a distinct profile of estimated scores on the PedsQL subscales. Following explorative LCAs of the three separate measurement moments, LTA was used to allow analysis of the same number of latent classes over time, as explored in the separate LCAs. LTA is a longitudinal extension of LCA over observed individuals' transitions in and out of latent classes, reflecting change in the latent variable of interest [20]. LTA estimates the proportion of individuals in each class at each time point, and the probability of transitioning from one class to another, conditioned on prior membership status.

In using LCA several statistical indicators can be used to select between models with different numbers of latent classes [21]. Model fit was assessed using the Akaike Information Criterion (AIC), Bayesian Information Criterion (BIC), Adjusted BIC, entropy and the Lo-Mendell-Rubin test (LMRT). The final LTA model was chosen on the basis of BIC and entropy, while we also required that adding an additional class should be clinically meaningful with a sample size sufficient for further statistical analyses. Following LTA, individuals were assigned to the latent class with the highest posterior probability across time.

In order to examine the construct validity of the latent classes, we compared the final latent classes with regard to well-being problems with mobility, self-care, usual activities, pain/discomfort and anxiety/depression, feeling happy at school and at home and probable depression caseness using multinomial logistic regression analyses (with the group with the highest QoL as reference category).

**Associated characteristics of QoL problems.**   We used multinomial logistic regression analyses to evaluate predictors of group membership (with the group with the highest QoL as reference category), while computing standard errors taking into account non-independence of observations due to cluster sampling using the Hubert-White procedure [22]. Unadjusted univariate and adjusted multivariable prediction models were constructed to estimate odds ratios (ORs plus 95% CIs) for the associations between predictors and group membership. All risk factors found to be significant in the univariate analyses were entered into the overall multivariable model. If two predictors were collinear (VIF > 2.0), they were investigated separately and the variable with most clinical relevance was retained.

**Predicting factors of persistent QoL problems.**   Similar fit logistic regression analyses accounting for non-independence of observations due to cluster sampling were used to analyze predictors of transitioning from a Low QoL group at baseline to a more adaptive QoL group at FU2. As longitudinal data were collected in the context of the CCG clustered RCT with four study arms, condition was coded with three dummy variables and entered as a covariate in order to examine the predictive value of risk factors for course independent of and above possible treatment effects.

Statistical analyses were conducted using the MPlus computer program (version 8) [22]. MPlus estimates a likelihood function for each individual based on the variables that are present so that all the available data are used (i.e., Full Information Maximum Likelihood (FIML) assuming missing data to be at least missing at random (MAR)).

**Ethics approval and consent to participate.**   The Scientific Ethics Review Unit at the Kenya Medical Research Institute (KEMRI), Nairobi (#3215) and the Liverpool School of Tropical Medicine (#15–005) reviewed the protocol, consent and assent documents, and questionnaires. Participation in the trial was entirely voluntary: girls had informed parent–guardian consent, gave their individual informed assent to participate and could withdraw from the study at any time. This study is reported as per the Strengthening the Reporting of Observational studies in Epidemiology (STROBE) guidelines (S1 Checklist).

## Results

### Participants

A total of 4,137 female students in secondary school class years Form 2 and 3 were enrolled between January 2017 and July 2018 across the 96 study schools. Complete baseline data and biomarkers were obtained for 3,998 girls and only girls with complete baseline data were included in analyzing associated characteristics of QoL problems and predictors of persistent QoL problems. Of these, 2,906 (72.7%) completed the PedsQL at FU1 and 3,275 (81.9%) at FU2. At FU2, 430 girls (13.1%) did not complete the subscale for school functioning because

items related to school were no longer applicable to them. Mean scores on the PedsQL varied across the specific subscales and diminished over time (S1 Table).

## Preliminary analyses

LCAs showed that a five-class solution was favoured over time with respect to the fit indices of AIC, BIC, and ABIC being the lowest, while the LMR-LRT test favoured a four-class solution at baseline and FU1 (S2 Table). Subsequently, analyzing the three-, four, and five-class model using LTAs showed that a four-class model had the lowest BIC while a five-class model failed to converge (S3 Table). We selected a three-class model as: (a) the class at the centre of the overall data distribution in the three-class model was split in two in the four-class solution, while the classes at the extremes of the overall data distribution remained almost unchanged; (b) our research questions pertained specifically to the latent class at the lower extreme of the data distribution; and (c) a four-class model resulted in latent classes of limited size not allowing meaningful transition analyses.

Based on the estimated means (Fig 1) we could classify the three different classes as: Low QoL, Average QoL and High QoL, as the estimated means for all dimensions of QoL were < 60 in the lowest group and > 90 in the highest group (see Huang [23] for clinically meaningful cut-off scores for the PedsQL). Classification quality as evaluated based on entropy was good (0.81 across all three time points) [24] (S3 Table) and average latent class probabilities for most likely latent class membership at all three time points were good (> .80) to excellent (>.90) (S4 Table). These classification quality indices supported the post hoc assignment of girls to transition classes based on their highest classification probability value.

## Prevalence of QoL and well-being problems

At baseline 1326 (28.2%) of the girls belonged to the Low QoL group, 1445 (36.1%) to the Average QoL group and 1427 (35.7%) to the High QoL group. In order to examine the

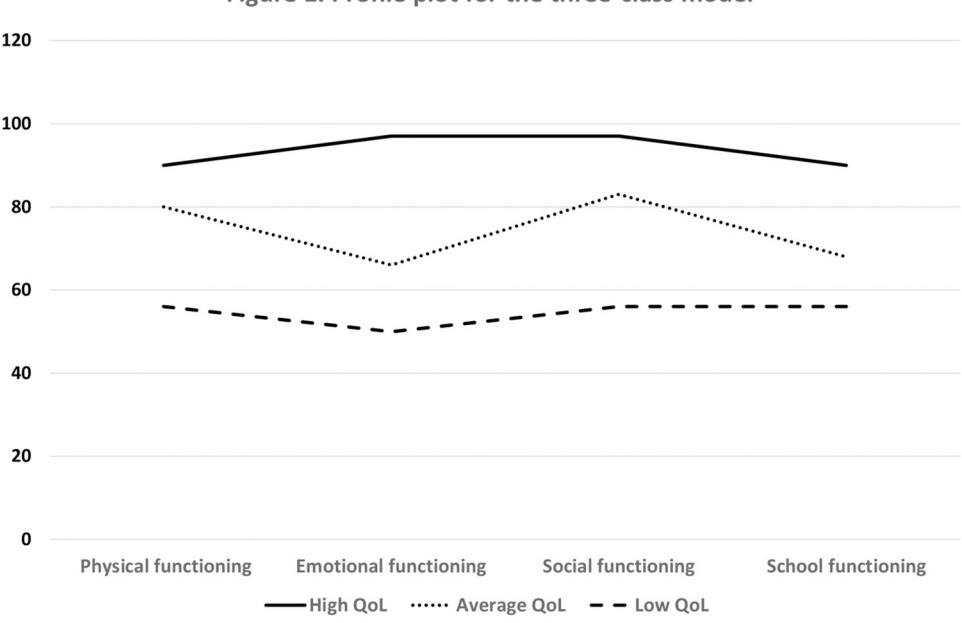

**Figure 1. Profile plot for the three-class model**

**Fig 1. Profile plot for the three-class model.**

**Table 1. Concurrent validity of the Low, Average and High QoL groups in relation to EQ-5D-3L, PHQ-9 and self-reported happiness at baseline (n = 3998).**

| | High QoL (n = 1427) | Av. QoL (n = 1445) | Low QoL (n = 1126) | Low QoL versus High QoL | Av. QoL versus High QoL |
|---|---|---|---|---|---|
| | N (%) | N (%) | N (%) | OR (95%CI) | OR (95%CI) |
| **EQ-5D-3L** | | | | | |
| Mobility problems | 136 (9.5) | 349 (24.2) | 483 (42.9) | 7.13 (5.92–8.58) *** | 3.02 (2.50–3.66) *** |
| Self-care problems | 107 (7.5) | 214 (14.8) | 303 (26.9) | 4.54 (3.57–5.78) *** | 2.12 (1.67–2.75) *** |
| Usual activities problems | 144 (10.1) | 367 (25.4) | 504 (44.8) | 7.22 (5.78–9.02) *** | 3.03 (2.48–3.71) *** |
| Pain/distress problems | 341 (23.9) | 60 (52.6) | 755 (67.1) | 6.48 (5.39–7.79) *** | 3.53 (3.01–4.15) *** |
| Anxiety/depression problems | 249 (17.4) | 571 (39.5) | 645 (57.3) | 6.34 (5.18–7.76) *** | 3.09 (2.58–3.70) *** |
| **PHQ9 case** (at FU2: n = 3275) | 37 (1.8) | 76 (9.1) | 59 (17.6) | 11.88 (7.96–17.74) *** | 5.56 (3.75–8.25) *** |
| **Unhappy at home** | 56 (3.9) | 93 (6.4) | 134 (11.9) | 3.31 (2.41–4.54) *** | 1.68 (1.19–2.38) ** |
| **Unhappy at school** | 22 (1.5) | 31 (2.1) | 61 (5.4) | 3.66 (2.21–6.05) *** | 1.40 (0.76–2.58) |

Note. QoL = Quality of Life; EQ-5D-3L = EuroQol-5 dimensions-3 levels (cutoff 1 vs 2/3); PHQ-9 = Patient health Questionnaire-9 (cutoff 0–9 vs 10–27)

*** < .001

** < .01.

construct validity of the three QoL groups into more detail, separate multinomial logistic regression analyses were executed with group membership as the dependent categorical variable and the High QoL group as the reference category. Predictors were the five items of the EQ-5D-3L (1 vs 2/3), self-reports of being unhappy at home ('no') and being unhappy at school ('no') at baseline and the PHQ-9 (with 10 as cut-off score [25]) at FU2 (Table 1). All predictors significantly predicted membership of the Low QoL group with ORs varying from 3.31 for unhappy at home to 11.88 for PHQ9 defined depression caseness. while all predictors also predicted membership of the Average QoL group -except unhappy at school—with intermediate OR values (Table 1). These results confirmed the expected intertwinement of quality of life with well-being problems.

## Associated characteristics of QoL problems

A next step in the analyses was to evaluate differences in sociodemographic, school, financial, general and reproductive health variables and adverse adolescent experiences between the three QoL groups. Table 2 presents the characteristics of the three groups and Table 3 shows the results of the multinomial logistic regression analyses in which group membership was regressed upon risk factors at baseline with the High QoL group (n = 1427) as reference category. The unadjusted models showed that higher age, lower SES, higher BMI, missing school, receiving money from a partner or working, various adverse sexual, physical and emotional adolescent experiences, as well as menstruation problems were more prominent in the low QoL group (n = 1126), while the average QoL group (n = 1445) also showed significantly higher intermediate scores for missing school, receiving money from a partner or working, adverse adolescent experiences and menstruation problems.

The adjusted predictive power of the risk factors was determined while controlling for all other significant risk factors. As noted in Table 3 in the adjusted models, receiving money from a partner (aOR = 1.66, p = .021), adverse adolescent experiences (sexual harassment out of school (aOR = 2.17, p < .001), indecent touching (aOR = 1.38, p = .011), physical assault (aOR = 1.53, p < .001), being robbed (aOR = 1.44, p = .003), threats to hurt you (aOR = 1.63, p < .001), threats for family to be hurt (aOR = 1.35, p = .008), being publicly humiliated (aOR = 1.43, p = .007), and menstruation problems (unavailability of pads (aOR = 1.23, p = .008), stopping activities (aOR = 1.77, p < .001) or missing school because of menstruation

**Table 2. Descriptives for associated characteristics of the different Quality of Life (QoL) Groups at baseline.**

| Associated characteristics | High QoL Group (n = 1427) | Average QoL Group (n = 1445) | Low QoL Group (n = 1126) |
|---|---|---|---|
| | N (%) | N (%) | N (%) |
| **Sociodemographics** | | | |
| Age categorical (year) (n = 3975) | | | |
| <16 | 285 (20.1) | 283 (19.7) | 191 (17.1) |
| 16 | 404 (28.5) | 421 (29.3) | 309 (27.7) |
| 17 | 429 (30.2) | 391 (27.2) | 304 (27.2) |
| 18 | 201 (14.2) | 212 (14.7) | 199 (17.8) |
| 19+ | 101 (7.1) | 131 (9.1) | 114 (10.2) |
| SES (poorest) | 559 (39.2) | 602 (41.7) | 546 (48.5) |
| Marital status (MCW) | 84 (5.9) | 102 (7.1) | 76 (6.7) |
| Baby at home to care for | 57 (4.0) | 61 (4.2) | 49 (4.4) |
| Orphan | 55 (3.9) | 44 (3.0) | 36 (3.2) |
| **School and Finances** | | | |
| Missed school—all reasons | 120 (8.4) | 216 (14.9) | 268 (23.8) |
| Missed school due to menstruation | 91 (6.4) | 160 (11.1) | 239 (21.2) |
| Received money from boyfriend/partner | 35 (2.5) | 72 (5.0) | 95 (8.4) |
| Received money from working | 144 (10.1) | 201 (13.9) | 215 (19.1) |
| **General Health** | | | |
| BMI categorical | | | |
| Underweight (BMI<18.2) | 77 (5.4) | 78 (5.4) | 57 (5.1) |
| Normal (BMI 18.2–25) | 1136 (79.6) | 1127 (78.0) | 863 (76.6) |
| Overweight (BMI>25) | 214 (15.0) | 240 (16.6) | 206 (18.3) |
| Drinking | 3 (0.2) | 5 (0.3) | 7 (0.6) |
| Smoking | 5 (0.4) | 2 (0.1) | 1 (0.1) |
| **Adverse Adolescent Experiences** | | | |
| Harassment for sex at school | 94 (6.6) | 180 (12.5) | 199 (17.7) |
| Harassment for sex out of school | 373 (26.1) | 642 (44.4) | 632 (56.1) |
| Touched indecently | 128 (9.0) | 218 (15.1) | 236 (21.0) |
| Sexually active | 293 (20.5) | 408 (28.2) | 389 (34.5) |
| Forced sex | 143 (10.0) | 214 (14.8) | 237 (21.0) |
| Physical assault | 305 (21.4) | 515 (35.6) | 499 (44.3) |
| Robbed | 163 ((1.4) | 266 (18.4) | 284 (25.2) |
| Threatened to hurt you | 200 (14.0) | 377 (26.1) | 429 (38.1) |
| Threats for family to be hurt | 255 (17.9) | 406 (28.1) | 447 (39.7) |
| Humiliation | 154 (10.8) | 311 (21.5) | 335 (29.8) |
| **Reproductive Health** | | | |
| HIV seropositive | 28 (2.0) | 19 (1.3) | 19 (1.7) |
| HSV-2 seropositive | 249 (17.4) | 246 (17.0) | 191 (17.0) |
| Early menarche <13 years) | 71 (5.0) | 90 (6.2) | 68 (6.0) |
| History of pregnancy (n = 1085) | 37 (12.7) | 51 (12.6) | 45 (11.6) |
| Menstruation severity | | | |
| Light | 123 (8.6) | 100 (6.9) | 87 (7.7) |
| Normal | 1055 (73.9) | 1036 (71.7) | 725 (64.4) |
| Heavy | 249 (17.4) | 309 (21.4) | 314 (27.9) |
| Menstruation duration (n = 3918) | | | |

*(Continued)*

**Table 2.** (Continued)

| Associated characteristics | High QoL Group (n = 1427) | Average QoL Group (n = 1445) | Low QoL Group (n = 1126) |
|---|---|---|---|
| | N (%) | N (%) | N (%) |
| <3 days | 48 (3.4) | 42 (3.0) | 43 (3.9) |
| 3–5 days | 1136 (81.1) | 1138 (80.4) | 845 (79.6) |
| >5 days | 217 (15.5) | 235 (16.6) | 214 (17.0) |
| Menstruation stopped activities | 258 (18.1) | 378 (26.2) | 460 (40.9) |
| No sanitary pads | 512 (35.9) | 544 (37.6) | 525 (46.6) |
| Had to do something to get sanitary pads | 123 (8.6) | 201 (13.9) | 242 (21.5) |

Note. BMI = Body Mass Index; SES = Socio-Economic Status; MCW/SO = married, cohabitating, widowed versus single/other.

problems (aOR = 1.77, p = .001), and having to do things to get sanitary pads (aOR = 1.44, p = .007) remained significant characteristics of the low QoL group. In the group with average QoL, sexual harassment out of school (aOR = 1.73, p < .001), physical assault (aOR = 1.43, p < .001), threats to hurt you (aOR = 1.31, p = .029), and being publicly humiliated (aOR = 1.38, p = .006) remained significant characteristics.

## Predicting factors of persistent QoL problems

For 220 (19.5%) of the 1126 girls from the Low QoL group at baseline no FU2 Peds-QL data were available. $\chi^2$-analyses revealed no significant differences between girls with and without FU2 data on any of the baseline covariates (except for alcohol use (0.3% vs. 1.8%, p = .01) and being sexually active (33.1% vs. 40.5%, p = .04)). In the remaining 906 girls from the Low QoL group, 415 (45.8%) transitioned to the Average QoL group and 285 (31.5%) to the High QoL group), while 206 (22.7%) remained in the Low QoL group. A chi-square analysis showed that the rate of transition was comparable between the treatment conditions of the CCG trial with masked labels for provision of cash, cups, both, or no intervention ($\chi^2$ (3) = 2.79, p = .43) allowing to examine predictors of improvement in the total Low QoL subgroup.

Table 4 presents the descriptive characteristics of the two course groups and shows the results of the logistic regression analyses in which persistence of QoL problems was regressed upon risk factors at baseline with the improved group as reference category. The unadjusted models showed that receiving money from working, sexual harassment at school, being sexually active, forced sex, physical assault, being robbed, threats to hurt you, threats for family to be hurt, being publicly humiliated, stopping activities because of monthly menstruation problems and having to do something to get sanitary pads were predictive of persistent QoL problems. In the adjusted models forced sex (aOR = 1.56, p = .028) and threats for family to be hurt (aOR = 1.98, p < .001) predicted persistence of QoL problems.

## Sensitivity analyses

In addition to the follow-up 'completers' analysis among girls with available FU2 PedsQL data described above, we repeated these analyses in the total group of girls from the Low QoL group, including those with missing PedsQL FU2 data (n = 220; 19.5%). According to the estimated likelihood function in LTA using FIML, from the 1126 girls from the Low QoL group, 20.5% remained in the Low QoL group (compared to 22.7% in the 'completers' analysis). In

**Table 3. Odds Ratio's (95% confidence intervals) for significant associated characteristics of Average QoL and Low QoL group at baseline (reference group High QoL group).**

| Associated characteristics | Average QoL group (n = 1445) | | Low QoL group (n = 1126) | |
|---|---|---|---|---|
| | Unadjusted model | Adjusted model | Unadjusted model | Adjusted model |
| | OR (95% CI) | OR (95% CI) | OR (95% CI) | OR (95% CI) |
| **Sociodemographics** | | | | |
| Age categorical (year) (n = 3975) | 1.03 (.96–1.11) | 1.00 (.94–1.07) | 1.12 (1.05–1.20) *** | 1.04 (.96–1.25) |
| SES (poorest vs less poor) | 1.10 (.96–1.290 | 1.02 (.87–1.200 | 1.46 (1.24–1.730 *** | 1.19 (.99–1.42) |
| **School and Finances** | | | | |
| Missed school—all reasons | 1.91 (1.51–2.43) *** | [a] | 3.40 (2.68–4.32) *** | [a] |
| Missed school due to menstruation | 1.83 (1.38–2.42) *** | 1.22 (.86–1.72) | 3.96 (3.06–5.12) *** | 1.72 (1.23–2.39) *** |
| Received money from boyfriend/partner | 2.09 (1.35–3.22) *** | 1.38 (.87–2.20) | 3.66 (2.52–5.33) *** | 1.66 (1.08–2.57) * |
| Received money from working | 1.44 (1.15–1.80) *** | 1.11 (.88–1.40) | 2.10 (1.60–2.76) *** | 1.26 (.95–1.66) |
| **General Health** | | | | |
| BMI categorical | 1.08 (.92–1.27) | 1.01 (.85–1.19) | 1.19 (1.05–1.35) ** | 1.06 (.91–1.23) |
| **Adverse Adolescent Experiences** | | | | |
| Harassment for sex at school | 2.02 (1.50–2.70) *** | 1.16 (.86–1.56) | 3.04 (2.35–3.94) *** | 1.24 (.92–1.66) |
| Harassment for sex out of school | 2.26 (1.93–2.64) *** | 1.73 (1.46–2.06) *** | 3.61 (3.08–4.24) *** | 2.17 (1.79–2.64) *** |
| Touched indecently | 1.80 (1.46–2.23) *** | 1.24 (.98–1.57) | 2.69 (2.21–3.28) *** | 1.38 (1.08–1.78) * |
| Sexually active | 1.52 (1.29–1.80) *** | [b] | 2.04 (1.69–2.47) *** | [b] |
| Forced sex | 1.56 (1.27–1.92) *** | .81 (.65–1.03) | 2.39 (1.93–2.96) *** | .82 (.63–1.07) |
| Physical assault | 2.04 (1.72–2.41) *** | 1.43 (1.20–1.70) *** | 2.93 (2.44–3.50) *** | 1.53 (1.24–1.88) *** |
| Robbed | 1.75 (1.40–2.18) *** | 1.23 (.97–1.57) | 2.62 (2.03–3.36) *** | 1.44 (1.10–1.89) ** |
| Threatened to hurt you | 2.17 (1.76–2.66) *** | 1.31 (1.03–1.68) * | 3.78 (3.16–4.51) *** | 1.63 (1.30–2.04) *** |
| Threats for family to be hurt | 1.80 (1.51–2.14) *** | 1.11 (.91–1.35) | 3.03 (2.50–3.65) *** | 1.35 (1.08–1.68) ** |
| Humiliation | 2.27 (1.84–2.82) *** | 1.38 (1.10–1.73) ** | 3.50 (2.79–4.39) *** | 1.43 (1.10–1.86) ** |
| **Reproductive Health** | | | | |
| Menstruation severity | 1.23 (1.09–1.39) *** | 1.08 (.95–1.23) | 1.51 (1.31–1.74) *** | 1.08 (.93–1.26) |
| Menstruation stopped activities | 1.60 (1.34–1.92) *** | 1.21 (.99–1.49) | 3.13 (2.58–3.79) *** | 1.77 (1.41–2.24) *** |
| No sanitary pads | 1.08 (.91–1.28) | .99 (.83–1.17) | 1.56 (1.34–1.82) *** | 1.23 (1.05–1.44) ** |
| Had to do something to get sanitary pads | 1.71 (1.30–2.26) *** | 1.29 (.96–1.73) | 2.90 (2.29–3.67) *** | 1.44 (1.11–1.88) ** |

Note.

[a] Excluded because of multicollinearity with 'Missed school due to menstruation'

[b] Excluded because of multicollinearity with 'Forced sex'; BMI = Body Mass Index; SES = Socio-Economic Status

*** < .001

** < .01

* < .05.

contrast to the 'completers' analysis, BMI was no longer a significant risk factor for persistent QoL problems, while the other significant predictors were the same as in the completers analysis (S5 Table).

We then repeated these analyses in the 794 girls from the Low QoL group with complete PedsQL data (i.e., also scores on the School functioning subscale). Among these girls 24.1% remained in the Low QoL group. Logistic regression models revealed that in addition to forced sex (aOR = 1.65, p = .021) and threats for family to be hurt (aOR = 1.92, p < .001), also orphanhood (aOR = 3.17, p = .015) predicted persistence of QoL problems in the adjusted models (S6 Table).

**Table 4.** Descriptives and Odds Ratio's (95% confidence intervals) for significant risk factors for remaining in Low QoL Group (analysis among 'completers': n = 906) (2 years after baseline).

| | Improved (n = 700) | Not-improved (n = 206) | Not-improved versus Improved | |
|---|---|---|---|---|
| | N (%) | N (%) | OR (95% CI) | aOR (95% CI) |
| **Risk factors** | | | | |
| **School and finances** | | | | |
| Received money from working | 124 (17.7) | 52 (25.2) | 1.57 (1.08–2.26) * | 1.29 (.85–1.97) |
| **General Health** | | | | |
| BMI categorical | | | 1.39 (1.04–1.87) * | 1.28 (.93–1.76) |
| Underweight (BMI<18.2) | 44 (6.3) | 7 (3.4) | | |
| Normal (BMI 18.2–25) | 540 (77.1) | 156 (75.7) | | |
| Overweight (BMI>25) | 116 (16.6) | 43 (20.9) | | |
| **Adverse Adolescent Experiences** | | | | |
| Harassment for sex at school | 109 (15.6) | 49 (23.8) | 1.69 (1.09–2.63) * | 1.25 (.77–2.01) |
| Sexually active | 211 (30.1) | 89 (43.2) | 1.76 (1.30–2.39) | a |
| Forced sex | 123 (17.6) | 67 (32.5) | 2.26 (1.58–3.24) *** | 1.56 (1.05–2.32) * |
| Physical assault | 282 (40.3) | 114 (55.3) | 1.84 (1.35–2.50) *** | 1.17 (.78–1.74) |
| Robbed | 156 (22.3) | 65 (31.6) | 1.61 (1.13–2.29) ** | 1.17 (.77–1.79) |
| Threatened to hurt you | 233 (33.3) | 103 (50.0) | 2.00 (1.51–2.66) *** | 1.01 (.67–1.51) |
| Threats for family to be hurt | 239 (34.1) | 118 (57.3) | 2.59 (1.90–3.52) *** | 1.98 (1.38–2.82) *** |
| Humiliation | 182 (26.0) | 88 (42.7) | 2.12 (1.51–2.98) *** | 1.42 (.96–2.09) |
| **Reproductive Health** | | | | |
| Menstruation stopped activities | 263 (37.6) | 101 (49.0) | 1.59 (1.16–2.21) ** | 1.24 (.86–1.80) |
| Had to do something to get sanitary pads | 146 (20.9) | 58 (28.2) | 1.49 (1.05–2.10) * | 1.02 (.70–1.49) |

Note. [a] Excluded because of multicollinearity with 'Forced sex'.

## Discussion

Our first research question was to determine the extent of QoL problems among adolescent secondary schoolgirls in rural Kenya. In the total sample, the mean level of QoL proved to be well within the range of those from large standardization samples in the United States [26]. These results concur with those of two previous variable-centered studies among sub-Saharan adolescents [12,13], also showing that overall the adolescent-reported mean level of QoL is relatively high. However, using a person-centered approach, we were able to show that mean scores for the whole sample may be misleading as we could identify a latent class with 28% of girls reporting substantially lower levels of QoL.

Previous studies have found that QoL is negatively related to both the presence and the severity mood-related disorders [27], and PEDS QoL scores have been found to reflect changes in depression severity [28]. Also, studies among sub-Saharan adolescents reported that QoL is affected by mental health issues [12,13]. Similarly, girls from our lower QoL group reported psychological health issues in particular (such as a higher prevalence of probable severe depression, not feeling happy at home or school, feelings of pain/distress, and anxiety/depression). As QoL and mental health problems are intrinsically intertwined, mental health practitioners in Kenya can use QoL scores to gauge the potential effects of problems on the mental well-being of children and youth referred to their practice [12].

The 28 percent of schoolgirls in the Low QoL group falls well within the range of prevalence rates of child and adolescent common mental health problems in sub-Saharan Africa as reported in previous studies among normative samples [4].

Our second research question was to examine the associated characteristic of girls with low QoL. As the present study was embedded in a trial focusing on SRH issues, we were able to show in more detail than in previous studies that particular aspects of SRH determine QoL. Girls from the Low QoL group were characterized by various adverse adolescent experiences such as exposure to sexual, physical, and emotional abuse. Previous studies have shown a higher prevalence of child sexual abuse in sub-Saharan Africa compared to most other regions in the world, with higher prevalence rates among girls than boys [29]. In African secondary high school children, lifetime exposure to sexual violence was reported by 9–33% (average: 23%) [30]. In addition to exposure to sexual violence, exposure to physical violence is also prevalent among children in sub-Saharan countries, with rates for physical exposure during the last 12 months varying from 27–50% (average: 42%) [30]. In the Low QoL group in our study, 21.0% reported forced sex, 44.3% physical assault, and 29.8% public humiliation. These rates are comparable to those reported in the most recent second Violence Against Children and Youth Survey (VACS) led by the Government of Kenya in 2019: 25.2% for any sexual violence, 45.9% for any physical violence, and 16.8% for any emotional violence in girls aged 13–24 years [31].

Reproductive health issues constitute an additional and severe problem facing menstruating adolescent schoolgirls. The unavailability of sanitary pads (46.6%), having to do something to obtain sanitary pads (21.5%) and stopping activities (40.9%), or missing school because of menstrual problems (21.2%) were additional relevant associated characteristics of girls from the low QoL group (controlling for all other significant baseline predictors). Relatedly, receiving money from a partner remained a significant predictor of low QoL in multivariate analyses. While mental health outcomes associated with inadequate menstrual hygiene were considered a research priority [32], no menstrual-related studies have evaluated mental health associated with poor menstrual care due to limited questioning using mental health scales. A lack of products for menstrual hygiene management, awareness, and facilities, as well as stigma, are pervasive problems preventing good management of menstruation in LMIC [33]. The lack of money to buy sanitary products may aggravate these problems. The present study clearly shows that these combined reproductive health and financial problems may contribute to a low QoL in adolescent menstruating girls.

In line with previous research [4,5] we also identified other associated characteristics only significant in univariate analyses, such as older age, lower SES and higher BMI, while other factors frequently mentioned in the literature (such as being married, HIV, alcohol abuse) were not characteristic of girls with a low QoL in our sample possibly because of their low prevalence.

Our third research question was to identify risk factors predictive of the persistence of QoL problems. To our knowledge, this is one of the first studies looking longitudinally at predictors of persistence of QoL problems in sub-Saharan adolescents allowing tentative directional inferences about the direction of the relationship of baseline factors with the course of QoL problems. Multiple adverse adolescent experiences (i.e., sexual harassment at school, being sexually active, forced sex, physical assault, being robbed, threatened to be hurt or family to be hurt, humiliation), reproductive health issues (stopping activities because of menstrual problems and having to do something to get sanitary pads), factors related to financial resources (i.e., receiving money from working) and BMI proved to be influential in predicting the persistence of QoL problems. In the adjusted models, forced sex (32.5% in not-improved vs. 17.6% in improved girls) and threats for family to be hurt (57.3% in not-improved vs. 34.1% in improved girls) remained significant predictors of persistent QoL problems.

Overall, these results concur with those of previous studies in high-income countries showing that adverse childhood and adolescent experiences contribute to mortality and morbidity

and have long-lasting effects on mental health, as well as drug and alcohol misuse, risky sexual behaviour, obesity, and criminal behaviour, which persist into adulthood [34]. Moreover, exposure to multiple adverse experiences during childhood and adolescence results in a much higher risk for sexual risk-taking, mental illness, problematic alcohol, and drug use, and interpersonal and self-directed violence [35]. Little is known about the long-term impact of these adverse experiences on health outcomes in low-income, high-violence settings, where exposure to adversity is common across the life-course. Extant limited research in a sample of males and females aged 15–26 years however shows that also in these settings adverse childhood and adolescent experiences are prospectively associated mental health problems, as well as with substance abuse and HIV risk [36].

Since the first 2010 VACS, Kenya has adopted a multisectoral approach to address adverse childhood and adolescent experiences involving relevant government departments and institutions, civil society organizations, and bilateral partners. This policy has helped to reduce the prevalence of violence among young adults [31]. However, the rate of physical, sexual, and emotional abuse remains high and constitutes an important global public health and human rights issue [37]. The most frequently used primary prevention strategy involves universal educational programs generally delivered in schools. This universal approach has many advantages as they can be offered at low cost, are easy to implement widely, and reach a maximum number of children while avoiding stigmatizing a particular population. However, this approach places the responsibility for the prevention of a complex social problem unduly in the hands of children and adolescents. A multi-factorial approach to reduce the rate of adverse childhood and adolescent experiences is needed, targeting not only personal but also family [38] as well as societal norms that influence the risk of assault [39].

## Study limitations

Our results should be interpreted in light of several limitations. Our findings may not be generalizable to all adolescent girls in these settings as our inclusion criteria and selection of menstruating public school day scholars followed the trial design. We cannot extrapolate results to the experience of boarding schoolgirls, who likely have lower exposure to sexual and reproductive harms in community settings, or to out-of-school girls who may have higher exposure risks, and potentially greater vulnerability to develop QoL and mental health problems. Among our study girls with low QoL the attrition rate was 19.5%, but only scarce evidence for selective attrition was found and the results of the analyses among girls with complete, incomplete and estimated FU2 PedsQL data proved to be very comparable. As our study was embedded in a RCT, follow-up measurements may have been influenced by receiving an intervention. However, analyses blind to trial arm yielded no evidence that the persistence of QoL problems in the subgroup of girls from the Low QoL group at baseline was affected by trial arm.

The time frames for health behaviour questions varied regarding either the last six months preceding the survey or the entire lifetime. We also note that most of the data are self-reported. Some girls may e.g., have misreported their adverse experiences or mental health problems either out of embarrassment or to provide a socially desirable response. However, there is no reason to believe that girls would systematically misreport in a manner that reflects the observed associations. Misreporting is most likely non-differential and consequently biasing our results towards the null hypothesis.

Caution is needed in making cross-cultural comparisons based on standardized self-report scales such as the PedsQL as long as their cross-cultural measurement invariance is not firmly established in the local context of the study [40], but we note that internal consistency

estimates of the PedsQL and PHQ-9 in the present study were satisfactory to good. The present study constitutes a post hoc analysis of data collected as part of a larger trial with specific SRH objectives, and so all analyses presented here were not planned prior to survey development. Consequently, some of our quality of life and health behaviour variables are limited in their detail, restricting interpretation.

## Conclusion

Our findings indicate that a substantial group of adolescent girls attending school in rural Kenya have serious QoL problems associated with adverse physical and sexual experiences, as well as problems with coping with their menstruation. A multi-factorial integral approach to reduce the rate of adverse adolescent experiences is needed targeting not only personal, but also family as well as societal norms that influence the risk of assault. Moreover, providing products for menstrual hygiene management may enhance girls' quality of life by fostering autonomy and reducing the risk of exposure to assault and emotional abuse.

## Supporting information

**S1 Checklist. STROBE statement—checklist of items that should be included in reports of *cross-sectional studies.***
(DOCX)

**S1 Table. Means and standard deviations at baseline, FU1, and FU2 on the Pediatric Quality of Life inventory (PEDS-QL).** Note. [a] n = 2845 because school functioning school items at FU2 were no longer applicable to all respondents.
(DOCX)

**S2 Table. Results of latent class analysis at baseline, FU1 and FU2.** Note. LL = Log-likelihood, AIC = Akaike Information Criterion; BIC = Bayesian Information Criterion LMR-LRT = Lo-Mendel-Ruben Likelihood Ratio Test.
(DOCX)

**S3 Table. Results of the LTA across baseline, FU1 and FU2 (n = 3998).** Note. LL = Log-likelihood, AIC = Akaike Information Criterion; BIC = Bayesian Information Criterion.
(DOCX)

**S4 Table. Average latent class probabilities for the most likely latent class membership at baseline, FU1 and FU2 (n = 3398).**
(DOCX)

**S5 Table. Odds Ratio's (95% confidence intervals) for remaining in Low QoL Group (ITT analysis: n = 1126).** Note. [a] Excluded because of multicollinearity with 'Forced sex'; BMI = Body Mass Index; SES = Socio-Economic Status; MCW versus SO = married, cohabitating, widowed versus single/other.
(DOCX)

**S6 Table. Odds Ratio's (95% confidence intervals) for remaining in Low QoL Group (completers analysis: n = 794).** Note; [a] Excluded because of multicollinearity with 'Forced sex'; BMI = Body Mass Index; SES = Socio-Economic Status; MCW versus SO = married, cohabitating, widowed versus single/other.
(DOCX)

## Acknowledgments

We are grateful to the girls, schools, community, and the Ministry of Health and Education partners for their participation in this study. We would like to thank the field and office staff for their support. The KEMRI Director approved publication of this paper.

## Author Contributions

**Conceptualization:** Philip Spinhoven, Duolao Wang, Daniel Kwaro, Penelope A. Phillips-Howard.

**Data curation:** Garazi Zulaika, Anna Maria van Eijk, David Obor.

**Formal analysis:** Philip Spinhoven, Duolao Wang.

**Funding acquisition:** Duolao Wang, Daniel Kwaro, Penelope A. Phillips-Howard.

**Investigation:** Philip Spinhoven, Garazi Zulaika, Elizabeth Nyothach, Penelope A. Phillips-Howard.

**Methodology:** Garazi Zulaika, Anna Maria van Eijk, David Obor, Linda Mason, Duolao Wang, Daniel Kwaro, Penelope A. Phillips-Howard.

**Project administration:** Elizabeth Nyothach, Eunice Fwaya.

**Resources:** Elizabeth Nyothach, David Obor, Eunice Fwaya.

**Supervision:** Garazi Zulaika, Elizabeth Nyothach, Duolao Wang, Penelope A. Phillips-Howard.

**Validation:** Garazi Zulaika, Elizabeth Nyothach, Anna Maria van Eijk, David Obor, Eunice Fwaya, Linda Mason.

**Visualization:** Philip Spinhoven.

**Writing – original draft:** Philip Spinhoven.

**Writing – review & editing:** Philip Spinhoven, Garazi Zulaika, Elizabeth Nyothach, Anna Maria van Eijk, David Obor, Eunice Fwaya, Linda Mason, Duolao Wang, Daniel Kwaro, Penelope A. Phillips-Howard.

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
