## [Decision Letter · Decision Letter 0]

8 Aug 2022

PGPH-D-22-01029

Quality of life and well-being problems in secondary schoolgirls in Kenya: Prevalence, associated characteristics, and course predictors

Dear Dr. Phillips-Howard,

Thank you for submitting your manuscript to PLOS Global Public Health. After careful consideration, we feel that it has merit but does not fully meet PLOS Global Public Health’s publication criteria as it currently stands. Therefore, we invite you to submit a revised version of the manuscript that addresses the points raised during the review process.

Please address all the comments raised by the reviewers. 

We look forward to receiving your revised manuscript.

Kind regards,

Palash Chandra Banik, MPhil

Academic Editor

Journal Requirements:

1. Please state the approval or name of the committee/IRB in the manuscript.

3. We ask that a manuscript source file is provided at Revision. Please upload your manuscript file as a .doc, .docx, .rtf or .tex with pdf.

4. Please update the Funding Information in the system to reflect the same information as Financial Disclosure Statement.

Additional Editor Comments (if provided):

Reviewers' comments:

Reviewer's Responses to Questions

**Comments to the Author**

1. Does this manuscript meet PLOS Global Public Health’s publication criteria? Is the manuscript technically sound, and do the data support the conclusions? The manuscript must describe methodologically and ethically rigorous research with conclusions that are appropriately drawn based on the data presented.

Reviewer #1: Partly

Reviewer #2: Yes

Reviewer #3: No

2. Has the statistical analysis been performed appropriately and rigorously?

Reviewer #1: Yes

Reviewer #2: Yes

Reviewer #3: I don't know

3. Have the authors made all data underlying the findings in their manuscript fully available (please refer to the Data Availability Statement at the start of the manuscript PDF file)?

Reviewer #1: No

Reviewer #2: Yes

Reviewer #3: Yes

4. Is the manuscript presented in an intelligible fashion and written in standard English?

Reviewer #1: Yes

Reviewer #2: Yes

Reviewer #3: No

5. Review Comments to the Author

Reviewer #1: The article is written in a plain English language. The general objective/aim is presented at the end of the introduction. The “material and method” presents the research procedure, but with an unnecessarily detail. Results are presented in the tables and figures, which demonstrates n, %, OR and 95% CI. Additional information such as funding and competing interest are provided. The scales used are widely used elsewhere. Next I justify the answers to the review questions. There are some grammatical and spelling errors that needs to be addressed thoroughly in revised manuscript.

I. Regarding question 1, I replied “partly” because the tested hypotheses are not presented early in the introduction, nor it is presented elsewhere in the manuscript. Tested hypothesis should be presented early in the manuscript, so that the reader can analyze and compare results presented, accordingly.

II. Regarding question 3, the original dataset is not uploaded, however, the data points, behind means, medians, IQR, are available.

Some additional points to be considered while revising the manuscript is detailed below.

Introduction

1. It was better to write the median point prevalence rate (IQR) were … . the IQR is missing in the parenthesis for anxiety.

2. “Included studies in this and a previous systematic review”, it is not clear what does the word “this” refer to. Although one can imagine that “this” may refer to the reference number 4, but is not scientifically correct to start a paragraph with a demonstrative pronoun that came in the previous paragraph.

3. I believe references 6, 7, 8, 9, 10 and 11 could be combined and shortened. They reveal similar findings (girls showing higher rates of depressive syndromes than boys). Writing above references in detail made the introduction section too long.

4. The paragraph that started at the bottom of page 5 and continued to middle of page 6, is too clumsy, has grammatical errors, and a vague presentation of data. The author pointed to the QoL and presented a definition, but failed to provide a reference. At the end of this paragraph, authors compared the findings of reference 12 with another study, which is not a scientific practice, in the introduction.

5. The last paragraph of introduction starts with the phrase “in conclusion”. This is not a good place for this phrase, as the introduction section does not have a concluding statement.

6. The aim of the study could have been presented as “the aim of this study is to assess prevalence, associated characteristics and predicting factors of QoL and well-being problems among schoolgirls in Kenya.”

7. The hypothesis of the study is not presented in the introduction. It is a scientific practice to present hypothesis in the introduction, so that readers understand what they are looking for.

Materials and methods

8. In “study area and population”, in line 2, “in” is written twice.

9. In “Study recruitment and procedures” the word “ministry” is not necessary at the beginning of line 4.

10. In “Study recruitment and procedures” it is better to refer the reader to original reference and not include all details in the current manuscript. (Study samples were recruited as mentioned elsewhere [14]”.

11. In “Study recruitment and procedures” references 16, 17, and 18 are missing. Authors failed to provide references in a consecutive order. Reference 15 has been mentioned previously, in “Study area and population”; then authors suddenly jumped to reference 19, without mentioning above mentioned missing references.

12. In “Outcome measurements”, when reference 16 and 17 are provided, reader can refer to them and it is not necessary to provide details of the instruments again in the current manuscript. In fact, the essence of referencing is to avoid repetition of information between literature.

13. In “Outcome measurements”, it is better to provide zero before a decimal number (for example range 0.68 - 0.89).

14. In “risk factors”, line 3 “SES” should be written in full, in the first mention.  

15. Line 4, page 11, “observes” should be changed to “observed”.

Results

1. Figure 2 is labelled as figure 1, at the end of the manuscript.

2. Description of table 1 does not clearly demonstrate what actually is in the Table. The High QoL, Average QoL, and Low QoL were not clearly defined in Table 1. Cross-ref of the information and Table 1, is very difficult.

3. Many overlapping information and data are presented in the Results section, that actually belong to the Material and Methods section. These really need to be controlled and adjusted in the revised version.

4. The headings of table 3 could be modified as OR (95% CI), beneath which both columns could be joined to shrink the size of the table. For example, for baseline data, mobility problem it could be written 7.13 (5.92-8.58). this way data could be more readable, and clearly presented.

5. Table 4, it is better to combine N and % in one column as “N (%)”. This way, the table looks neater and more readable.

6. Table 4 is confusing, for example under the age categories (years), <16, 16 … looks to be in a separate column, with no heading. It is better to re-organize the table to be more readable. Moreover, presentation of data is not straightforward. For example, for “SES (poorest vs less poor)” one cannot understand the corresponding numbers and percentages are for those who are poorest, or those who are less poor; and for “Marital status (MCW/SO)” similar problem exists. It is also better to write the number of participants corresponding to each variable next to it, not in the footnote of the table.

7. Table 5 and Table 6, similar to Table 3, it is better to combine OR and 95% CI in the same column as OR (95% CI). This way, data will be presented neatly and more readably. I also believe that if authors used Maximum Likelihood ratio model, it was much better to show only those variables that were significantly predicting high QoL. Furthermore, the significance value (p value) is not provided, one cannot understand which factors are considered associated or predicting factors for QoL.

8. The result section, overall, is presented in a way that is not easy to understand, contains not so much relevant information, and overlapping data/information from the Material and Method section. If authors focused only on three components of their study question/aims, the Results was more structured and reader-friendly. I believe authors only needed to report 1) prevalence 2) associated factors, and 3) predicting factors of low QoL and well-being among participants. Other Tables, Figures, and Supplementary material were not required. Authors, I believe, mixed the two projects (CCG and current study question) together, which caused writing a long manuscript with information and data jumping from one matter to the next. I recommend that authors more focus on the study question and re-write the Results section only on prevalence, associated factors and predicting factors of low QoL and well-being. Much of the steps involved included in the Results section, are compulsory initial steps of a research, not required to report.

9. Figures are not readable and should be drawn afresh.

Discussion

1. Reference 26 refers to a study that was conducted in California, USA. But authors incorrectly referred to it as most high- and middle-income countries. One expects when the word “most” appears, authors should provide many references or at least a systematic review/meta-analysis study.

2. Authors wrote “QoL is negatively related to both the presence and the severity of psychopathology [27]”. But reference 27 refers to a study that only focused on “mood-related disorders”. Although mood-disorders are a type of psychopathology, it is not the only one. Therefore, it is better to clearly write ““QoL has been negatively related to both the presence and the severity of mood-related disorders [27]”.

3. In the last part of the first paragraph in Discussion, author concluded their findings, but also provided references. If it is what authors believe that came from their study, no reference is required. If it is similar to what others found and concluded, then this paper is not really “original”.

4. An incorrect use of writing signposts (for example, “Finally”) appears frequently in the discussion as well as introduction sections.

5. Grammatical error “As far as we are aware, is this one of the first studies looking longitudinally at predictors of persistence of QoL and well-being problems in sub-Saharan adolescents”.

6. The first paragraph on page 37 looks irrelevant to study subject.

7. Paragraph 2 and 3 on page 37 should go under “Limitation” heading.

8. In the last sentence of Discussion on page 38, authors indicated “Lastly, unobserved confounding precludes drawing strong causal inferences from observational data.” This means that even authors themselves do not rely on the results of their study to reveal any causal relationship between associated characteristics and low QoL and well-being.

Reviewer #2: Comments

Congratulations to the authors for the article on this very important topic and about which there is still much to understand.

Overall, the paper is well done, giving the fact that there is limited data in this area of research in Sub-Saharan Africa.

That notwithstanding, I have some few comments that needs to be addressed by authors.

First, the 3rd sentence under introduction section, “Adolescents living in low-and middle income countries (LMIC) are disproportionately affected, and the proportion of disease burden in children and youth attributable to mental and substance use disorders will increase as reproductive health and the management of infectious diseases improve in LMICs [3].” This statement doesn’t seem clear to me and authors need to review to maintain the intended meaning.

Secondary, could the authors explain why they opted to include only girls’ day schools and any special reason(s) for excluding boarding schools.

Reviewer #3: The essence of the manuscript is an important public health concern. However, there is need to rewrite the paper for a better reading and understanding. Statement like "have serious QOl and well-being is confusing and this appeared in several areas.

Also, the introduction, results, discussion and conclusion will need extensive revision

6. PLOS authors have the option to publish the peer review history of their article (what does this mean?). If published, this will include your full peer review and any attached files.

**Do you want your identity to be public for this peer review?** For information about this choice, including consent withdrawal, please see our Privacy Policy.

Reviewer #1: No

Reviewer #2: No

Reviewer #3: No

---

## [Decision Letter · Decision Letter 1]

9 Nov 2022

Quality of life and well-being problems in secondary schoolgirls in Kenya: Prevalence, associated characteristics, and course predictors

PGPH-D-22-01029R1

Dear Dr. Phillips-Howard,

We are pleased to inform you that your manuscript 'Quality of life and well-being problems in secondary schoolgirls in Kenya: Prevalence, associated characteristics, and course predictors' has been provisionally accepted for publication in PLOS Global Public Health.

Best regards,

Palash Chandra Banik, MPhil

Academic Editor

Reviewer Comments (if any, and for reference):

Reviewer's Responses to Questions

**Comments to the Author**

1. If the authors have adequately addressed your comments raised in a previous round of review and you feel that this manuscript is now acceptable for publication, you may indicate that here to bypass the “Comments to the Author” section, enter your conflict of interest statement in the “Confidential to Editor” section, and submit your "Accept" recommendation.

Reviewer #4: All comments have been addressed

2. Does this manuscript meet PLOS Global Public Health’s publication criteria? Is the manuscript technically sound, and do the data support the conclusions? The manuscript must describe methodologically and ethically rigorous research with conclusions that are appropriately drawn based on the data presented.

Reviewer #4: Yes

3. Has the statistical analysis been performed appropriately and rigorously?

Reviewer #4: Yes

4. Have the authors made all data underlying the findings in their manuscript fully available (please refer to the Data Availability Statement at the start of the manuscript PDF file)?

Reviewer #4: Yes

5. Is the manuscript presented in an intelligible fashion and written in standard English?

Reviewer #4: Yes

6. Review Comments to the Author

Reviewer #4: a good study

7. PLOS authors have the option to publish the peer review history of their article (what does this mean?). If published, this will include your full peer review and any attached files.

**Do you want your identity to be public for this peer review?** For information about this choice, including consent withdrawal, please see our Privacy Policy.

Reviewer #4: **Yes: **Olaf Jensen
